# Differences in splicing defects between the grey and white matter in myotonic dystrophy type 1 patients

**Masamitsu Nishi[1], Takashi Kimura[1]\*, Masataka Igeta[2], Mitsuru Furuta[3], Koichi Suenaga[1,4], Tsuyoshi Matsumura[5], Harutoshi Fujimura[5], Kenji Jinnai[6], Hiroo Yoshikawa[1]**

1 Department of Internal Medicine Division of Neurology, Hyogo College of Medicine, Nishinomiya, Japan, 2 Department of Biostatistics, Hyogo College of Medicine, Nishinomiya, Japan, 3 Department of Neurology, Japan Organization of Occupational Health and Safety Kan-sai Rosai Hospital, Osaka, Japan, 4 Department of Internal Medicine, Japan Self Defense Force Hanshin Hospital, Kawanishi, Japan, 5 Department of Neurology, National Hospital Organization Toneyama Hospital, Toyonaka, Japan, 6 Department of Neurology, National Hospital Organization Hyogo-Chuo Hospital, Sanda, Japan

\* kimura@hyo-med.ac.jp

## 

**Data Availability Statement:** All relevant data are within the manuscript and its Supporting Information files.

**Funding:** This work was supported by JSPS KAKENHI Grant Number JP18K07515. The funders

## Abstract

Myotonic dystrophy type 1 (DM1) is a multi-system disorder caused by CTG repeats in the *myotonic dystrophy protein kinase* (*DMPK*) gene. This leads to the sequestration of splicing factors such as muscleblind-like 1/2 (MBNL1/2) and aberrant splicing in the central nervous system. We investigated the splicing patterns of *MBNL1/2* and genes controlled by MBNL2 in several regions of the brain and between the grey matter (GM) and white matter (WM) in DM1 patients using RT-PCR. Compared with amyotrophic lateral sclerosis (ALS, as disease controls), the percentage of spliced-in parameter (PSI) for most of the examined exons were significantly altered in most of the brain regions of DM1 patients, except for the cerebellum. The splicing of many genes was differently regulated between the GM and WM in both DM1 and ALS. In 7 out of the 15 examined splicing events, the level of PSI change between DM1 and ALS was significantly higher in the GM than in the WM. The differences in alternative splicing between the GM and WM may be related to the effect of DM1 on the WM of the brain.

## Introduction

Myotonic dystrophy type 1 (DM1) is the most common form of muscular dystrophy in adults, affecting the skeletal muscle, heart, ocular lens, testis, and central nervous system (CNS). The CNS symptoms of DM1 can have a negative impact on a patient's quality of life [1]. DM1 is caused by the unstable expansion of CTG trinucleotide repeats in the 3' untranslated region (UTR) of the *myotonic dystrophy protein kinase* (*DMPK*) gene. These CTG repeats are transcribed to CUG repeats, leading to the formation of an RNA hairpin loop. These loops form foci in the nucleus and sequester splicing factors, such as the muscleblind-like (MBNL) proteins. The MBNL proteins control splicing; in the nucleus, MBNL binds to the YGCY motif of

had no role in study design, data collection and analysis, decision to publish, or preparation of the manuscript.

**Competing interests:** The authors have declared that no competing interests exist.

pre-mRNAs and controls the splicing of alternative exons in either an exon inclusion or exon exclusion direction, depending on whether the motif is located in the downstream or upstream intron for each exon, respectively [2]. We have previously detected several splicing defects in the brains of both *MBNL1/2* knockout mice and DM1 patients, and revealed that MBNL2 is a major splicing factor in the brain, while MBNL1 performs a similar function in the skeletal muscle [3, 4]. In addition, conditional *Mbnl1* and *Mbnl2* double-knockout mice showed greater splicing defects than either *Mbnl1* or *Mbnl2* knockout, indicating that the combined loss of MBNL1 and MBNL2 is necessary to exacerbate mis-splicing, since MBNL1 and MBNL2 compensate for each other [5, 6].

We have previously examined the mis-splicing in DM1 patients for each area of the brain and found that mis-splicing in the cerebellum is less apparent than the other areas [3]. We also demonstrated the different degrees of mis-splicing that occurs among cell layers of the cerebellum [7]. These results clearly showed the heterogeneity of the brain and show that detailed splicing analysis for each cell or region is essential to clarify the pathomechanisms of this disorder.

It has been reported that fetal splice isoforms increase in adult DM1 tissues as a result of MBNL1/2 sequestration, which regulates the splicing switch from fetal to adult [4, 8–11]. *MBNL2* splicing is developmentally regulated; exon 5 (54nt.) and exon 8 (95nt.) are included in the fetal brain, while both exons are excluded in the adult brain [12].

Using RNA-sequencing, Mills et al. showed that the grey matter (GM) and white matter (WM) have distinct transcriptome profiles, including alternative splicing (AS) [13]. Neuroimaging analysis in patients with DM1 revealed various changes with conventional MRI [14, 15], voxel-based morphometry (VBM) [16, 17], and diffusion-weighted imaging (DWI) tensor analysis [18–21]. Overall, the degree of change is greater in the WM than in the GM [17, 22].

Aberrant splicing of *MBNL1/2* in the brain of DM1 patients and mouse models has been previously reported [23–25]. Considering distinct regions are affected, as shown by neuroimaging, and the differences in splicing regulation between the GM and WM, it is important to determine how AS is controlled in GM and WM of a DM1 brain. In this study, we examined the splicing patterns of *MBNL1/2*, and the other genes controlled by MBNL2, among several brain regions (frontal and temporal lobes, hippocampus, the cerebellum), and between the GM and WM in DM1 patients.

## Materials and methods

### Ethics and written informed consent

This research was approved by the Ethics Committee of Hyogo College of Medicine (approval number: 93), and written informed consent was obtained from patients themselves or from their family members for an autopsy.

### MBNL1/2 and APP DNA sequence and primer design

We searched the DNA sequences for *MBNL1*, *MBNL2* and *APP* using the GENETYX® NCBI database. *MBNL1* has over 50 variants depending on the presence or absence of exons 5 (54 nt.), 6 (154 nt.), 7 (36 nt.), 8 (95 nt.), and 9 (64 nt.), or changes in the non-coding region. *MBNL2* has 10 splicing variants depending on the presence or absence of exons 5 (54 nt.), 7 (36 nt.), and 8 (95 nt.). *APP* has 3 variants depending on the presence or absence of exon 7 (168 nt.) and exon 8 (57 nt.). Using NCBI Primer-BLAST, we designed two sets of primers for each spliced gene; *MBNL1/2* exon 4 forward and exon 6 reverse primers for exon 5 splicing, and exon 6 forward and exon 9 reverse primers for exons 7 and 8 splicing. *APP* exon 6 forward and exon 9 reverse primers for exon 7 and 8 splicing. In addition, we examined splicing

**Table 1. RT-PCR primers for alternative splicing.**

| Genes (target exon) | Forward primer | Reverse primer |
|---|---|---|
| *ADD1* (exon 15 (34 nt.)) | GGACGAGGCTAGAGAACAGAAAGAAAAGA | TTGGGAAGCCGAGTGCTTCTGAA |
| *APP* (exon 7 (168 nt.), exon 8 (57 nt.)) | TCTGTGGAAGAGGTGGTTCG | TGGCCTCAAGCCTCTCTTTG |
| *CACNA1D* (exon 12 (60 nt.)) | CACAGAGAACGTCAGCGGT | TGAGTTTGGATTTTGAGATGGC |
| *CLASP2* (exon 23a (27 nt.), exon 23b (27 nt.)) | GCTGGCATGGGAAATGCCAAGGC | GCTCCGTGGTATCTTGCTTCTTTTT |
| *CSNK1D* (exon 9 (64 nt.)) | GATACCTCTCGCATGTCCACCTCACA | GCATTGTCTGCCCTTCACAGCAAT |
| *GRIN1* (exon 4 (63 nt.)) | GTCTACAGCTGGAACCACATC | TCCATCAGCAGGGCCGTCACG |
| *KCNMA1* (exon 27a (81 nt.)) | CGTTCACACCTCCAGGAATGGATAGAT | GTGAGGTACAGTTCTGTATCAGGGTCAT |
| *MAPT* (exon 2 (87 nt.)) | TACACCATGCACCAAGACCA | GTCTCCAATGCCTGCTTCTT |
| *MAPT* (exon 10 (93 nt.)) | ACTGAGAACCTGAAGCACCAG | CACTTGGAGGTCACCTTGCTC |
| *MBNL1* (exon 5 (54 nt.)) | TCAAGGCTGCCCAATACCAG | TGTTGGCTAGAGCCTGTTGG |
| *MBNL1* (exon 7 (36 nt.), exon 8 (95 nt.)) | ACCAACAGGCTCTAGCCAAC | GGCTAGTCAGATGTTCGGCA |
| *MBNL2* (exon 5 (54 nt.)) | AGGCCAAAATCAAAGCTGCG | GTGAGAGCCTGCTGGTAGTG |
| *MBNL2* (exon 7 (36 nt.), exon 8 (95 nt.)) | CACGCCGCGTTCATTCCAAC | TAGCATGCAGTTTGTGGCAA |
| *TANC2* (exon 22a (30 nt.)) | GCCATGATCGAGCACGTTGACTACAGT | CCTCTTCCATCAGCTTGCTCAACA |

patterns of other genes controlled by MBNL2: *ADD1* exon 15 (34 nt.), *CACNA1D* exon 11 (60 nt.), *CLASP2* exon 23a (27 nt.) or 23b (27nt.), *CSCNK1D* exon 9 (64 nt.), *GRIN1* exon 4 (63 nt.), *KCNMA1* exon 27a (81 nt.), *MAPT* exon 2 (87 nt.) and exon 10 (93 nt.), *TANC2* exon 22a (30 nt.), using the primers used in our previous report [3, 7] (Table 1).

## Human RNA and splicing analysis

We investigated brains of patients during autopsy, with RNA extracted from the brain of 6 patients with DM1 and 6 patients with amyotrophic lateral sclerosis (ALS, as disease controls). The clinical features of samples are summarized in Table 2 [26]. Two RNA samples of fetal brains (as fetal controls) were obtained commercially (Cat. No. 540157, Agilent Technologies, US; Cat. No. 636526, Takara Bio Inc. Japan). RNA was extracted from the frontal and temporal lobes, the hippocampus, and the cerebellum using the ISOGEN® reagent (Nippon Gene, Japan). The splicing patterns in the different brain regions and between the GM and WM.

**Table 2. Clinical features of samples.**

| Disease | Gender | Age | Onset age | Duration | Type of DM1 | Cognitive decline |
|---|---|---|---|---|---|---|
| DM1 | F | 58 | 38 | 20 years | Adult onset | Mild |
| | F | 66 | 42 | 24 years | Late onset | Mild |
| | M | 47 | 12 | 35 years | Juvenile onset | Severe |
| | M | 65 | 40 | 25 years | Adult onset | Mild |
| | M | 78 | 41 | 37 years | Late onset | Mild |
| | M | 57 | 18 | 39 years | Juvenile onset | Mild |
| ALS | F | 53 | N/A | N/A | N/A | Normal |
| | M | 69 | N/A | N/A | N/A | Normal |
| | M | 67 | N/A | N/A | N/A | Normal |
| | F | 73 | N/A | N/A | N/A | Normal |
| | F | 73 | N/A | N/A | N/A | Normal |
| | M | 70 | N/A | N/A | N/A | Normal |

Type of DM1; Juvenile onset (age at onset 10–20 years), Adult onset (age at onset 20–40 years), Late onset (age at onset > 40 years). The degree of cognitive decline was assessed on a three-point scale; Normal, Mild, and Severe.

**Table 3. Quantitative RT-PCR primers for confirming the separation between the GM and WM.**

| Genes | Forward primer | Reverse primer |
|---|---|---|
| *NEFH* | AGGTGAAGAGTGTCGGATTG | GAAGCGAGAAAGGAATTGGG |
| *MOG* | CCTCCACTTGGCCTGACCTT | ACCTCCATGCCTGTAGCGTT |
| *GAPDH* | CCATCACTGCCACCCAGAAGAC | CCATCACGCCACAGTTTCCC |

For GM and WM, the frontal lobe tissue was sliced into 20-μm thick sections using a Cryostat (Leica® CM1520, Germany). Before separation, one of the sections was stained using the Luxol Fast Blue Stain Kit (ScyTek Laboratories Inc., US), to identify the boundary between the GM and WM. Only unstained sections were used for RNA extraction. We manually separated each section into the GM and WM and extracted RNA using RNeasy® Plus Mini (QIAGEN®, Germany) for comparison between the GM and WM.

To ensure enrichment of the GM and WM, we used the primers for *NEFH* (higher expression in the GM than the WM), *MOG* (higher expression in the WM than the GM), and *GAPDH* (control), and examined this by quantitative real-time PCR (qPCR) using the PowerUp™ SYBR™ with the 7500 Real-Time PCR system (Applied Biosystems, US) (Table 3). qPCR was performed in triplicates [13]. The levels of *NEFH* and *MOG* mRNA expression were calculated by the $2^{-\Delta\Delta Ct}$ method, using *GAPDH* as an endogenous control, and the data were presented as the mean of triplicates.

All cDNA was synthesized using the extracted RNA and purchased RNA (1 μg of RNA was used for comparison among several brain regions; 10–400 ng of RNA was used for comparison between the GM and WM). Random hexamers with the SuperScript® III First-Strand Synthesis System (Invitrogen™, US) for reverse transcription PCR (RT-PCR) were used according to manufacturer's instructions (Applied Biosystems). cDNA was amplified using AmpliTaq Gold® 360 Master Mix (Applied Biosystems), with the initial denaturation set at 94˚C for 10 min, followed by 32 (for comparison among several brain regions) or 36 (for comparison between the GM and WM) amplification cycles at 94˚C for 30 s, 61˚C for 30 s, and 72˚C for 60 s. PCR products were analyzed with an Agilent 2100 Bioanalyzer (Agilent Technologies, US). The linearity of the PCR amplification at each cycle was verified with the DM1 and ALS samples. We used percent spliced-in (PSI) values, indicating inclusion ratio of an alternative exon [3].

## Statistical analysis

The Welch's t-test was performed for testing mean difference of PSI between ALS and DM1 with two-sided significant level of 5% for all comparisons. No adjustment of significant level to multiplicity was applied to retain maximum power in this study, as suggested by Saville [27]. Since some data showed skew distribution, we evaluated the impact of the assumption of the normality of the Welch's t-test via sensitivity analyses as following steps; First, we performed the Shapiro-Wilk test for testing the normality. Then if the test of normality was significant at least one group within a group comparison, the Wilcoxon rank sum test was applied to the group comparison. Only the p-values of the Welch's t-test were shown in Figures since we confirmed the consistent significant results between the Welch's t-test and the Wilcoxon rank sum test for the cases where the test of normality was significant (data not shown). We calculated the average level of PSI change (ΔPSI), which was obtained by subtracting the average of PSI of the ALS from the DM1. The summary statistics, 95% confidence interval based on the unpooled variance, and p-value calculated using Welch's t-test for the difference between average ΔPSI GM and average ΔPSI WM for each exon are provided. The confidence interval and

p-value were evaluated for the difference between average PSI DM1 (GM-WM) and average PSI control (GM-WM) because these differences were mathematically equivalent in our data, which had no missing within-subject GM and WM PSIs for each exon.

## Results

### Comparison among several brain regions

Compared with the ALS, PSI for *MBNL1* exon 5 were higher in all examined brain areas of DM1. The PSI for *MBNL1* exon 8 were significantly higher in the DM1 temporal lobe than in the ALS (Fig 1, S1 Fig). The result of *MBNL1* exon 5 splicing was similar to a previous report [23], and that of exon 8 was not reported. Compared with the ALS, PSI for *MBNL2* exon 5 were higher in all examined brain areas of DM1 and exon 8 were significantly higher in most brain areas of DM1, except for the cerebellum. Notably, for the frontal and temporal regions, and hippocampus of DM1, PSI for *MBNL1* and *MBNL2* exons 5 and exon 8 were highly variable compared with the ALS.

Compared with the ALS, PSI for *MAPT* exon 2 and exon 10 were significantly lower in most brain areas of DM1, except for the cerebellum. PSI for *GRIN1* exon 4 was significantly higher in the DM1 hippocampus than in the ALS. Other areas had no significant difference. PSI was very low in the fetal brain tissue (Fig 1, S2 Fig).

### Comparison between the GM and WM

Q-PCR analysis showed that the expression levels of *NEFH* was higher in the GM, while that of *MOG* was higher in the WM (Fig 2), confirming that samples were enriched for the GM and WM.

There were three patterns of AS in the fetal brain: 1) exon inclusion type (PSI ≥ 60%, *ADD1*, *CLASP2*, *MBNL1/2*), 2) exon exclusion type (PSI < 40%, *APP*, *CACNA1D*, *CSNK1D*, *GRIN1*, *KCNMA1*, *MAPT*) (Fig 3, S2 Fig), and 3) moderate type (40% ≤ PSI < 60%, *TANC2*).

We used ΔPSI to compare splicing defects between GM and WM. For example, average PSI of *MBNL1* for exon 5 were 57.43%, 24.33%, 25.06%, and 12.35% in the GM and WM of DM1, and the GM and WM of ALS, respectively. ΔPSI of the GM was 32.38%, while ΔPSI of the WM was 11.99%. In 7 (*MBNL1* exon 5, *MBNL2* exon 5, *MBNL2* exon 8, *CLASP2* exon 23a or 23b, *CACNA1D* exon 12, *CSNK1D* exon 9, *MAPT* exon 2), out of 15 examined splicing events, | ΔPSI | of the GM was significantly higher than that of the WM, suggesting that more splicing misregulation occurs in the GM (Fig 4).

PSI of *APP* exon 8 and *GRIN1* exon 4 showed no statistically significant difference between DM1 and the ALS (Fig 3, S2 Fig).

## Discussion

Comparisons among the brain regions revealed that PSI of almost all examined gene exons were significantly different in the DM1 patients compared with the ALS, except for the cerebellum. Comparison between the GM and WM revealed that the splicing of many genes was differently regulated between the GM and WM. The extent of splicing change between the DM1 and ALS was higher in the GM than in the WM.

According to our previous study, there were fewer alternative splicing defects in the DM1 cerebellum than in other brain regions [3]. We showed that the inclusion of *MBNL1* exon 5 and *MBNL2* exon 5 and exon 8 was higher in most brain areas, except the cerebellum for *MBNL2* exon 5 and exon 8. *MBNL1/2* exon 5 inclusion in the DM1 human brain had previously been reported [23, 24]; however, we are the first to demonstrate *MBNL2* exon 8 inclusion

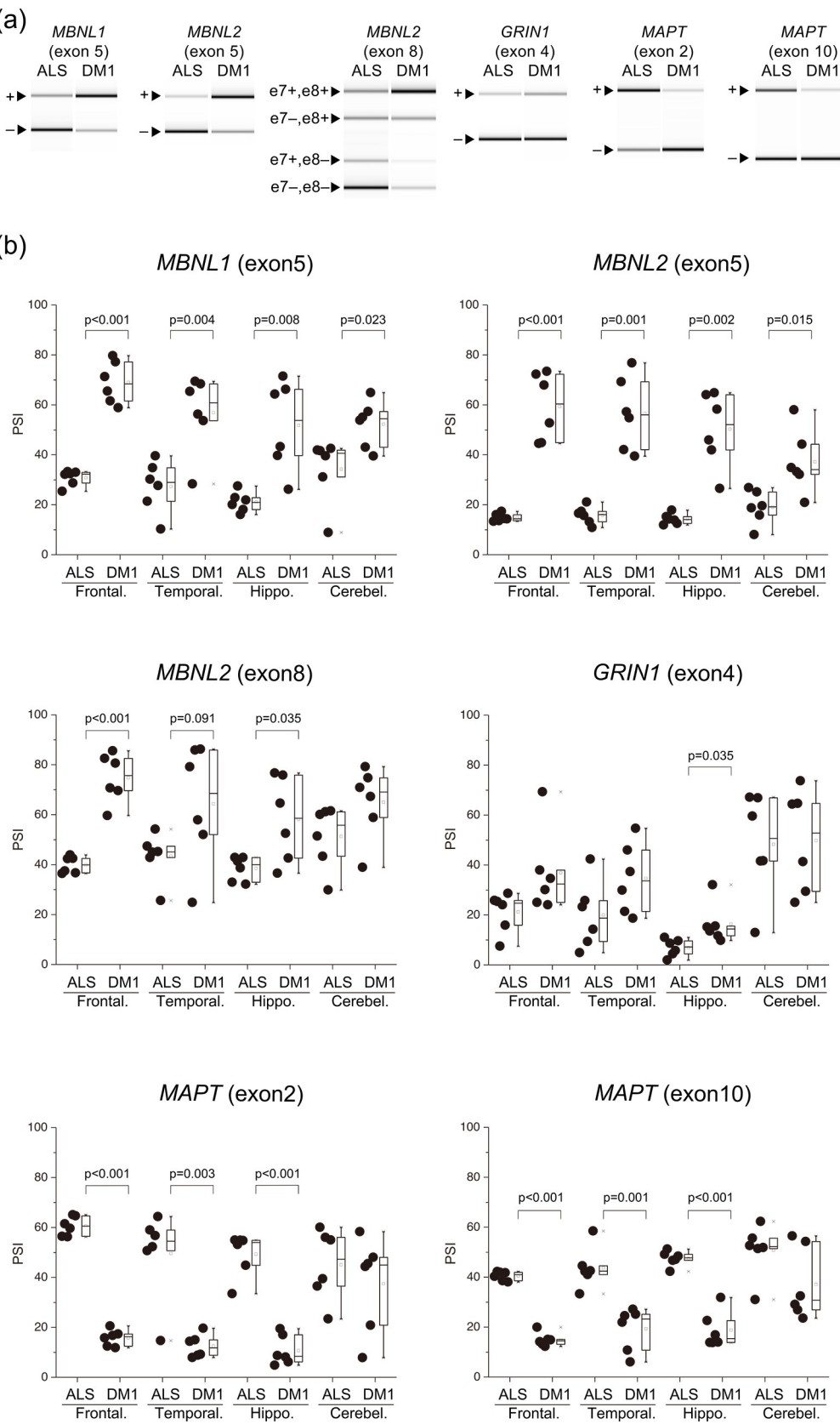

**Fig 1. Aberrant splicing among several regions of the brain.** (a) Representative RT-PCR products from the frontal lobe in the ALS and DM1. (b) Inclusion ratios of splicing changes in several brain regions. PSI values of all examined genes were compared by Welch's T-test. In the box-and-whisker plot, the line inside the box represents the median and the square symbol represents the average. ALS, amyotrophic lateral sclerosis; DM1, myotonic dystrophy type 1; Frontal., Frontal lobe; Temporal., Temporal lobe; Hippo., Hippocampus; Cerebel., Cerebellum.

in the DM1 human brain. *MBNL1/2* splicing in several brain regions were similar to previous reports [3], in that aberrant splicing is observed in most brain areas except for the cerebellum.

*MAPT* [28] and *GRIN1* [10] splicing defects were found in DM1 patients' brains. The aberrant splicing of *MAPT* exons 2, 3 [4], 10 and *GRIN1* [5] were recapitulated by using knock out mice of *Mbnl1/2*, suggesting that these splicing events are controlled by the MBNL 1/2 protein. We analyzed both these genes in several areas of the brain and found that *MAPT* splicing was similar to that of other genes, where there were fewer splicing changes in the cerebellum than in other brain areas in the DM1. *GRIN1* was different in that: 1) In the hippocampus, PSI of the DM1 was significantly higher than that of ALS. In the frontal and temporal lobes, PSI of the DM1 tended to be higher than that of the ALS, but these differences did not reach statistical significance; 2) PSI in the fetal brain was lower than that in the ALS, in contrast to the DM1. A significant difference was observed in the hippocampus between the ALS and DM1, but not in other areas. Interestingly, the splicing defect of *GRIN1* in the hippocampus does not represent a return to an embryonic splicing profile. However, ΔPSI is very low (~ 5%), and thus the

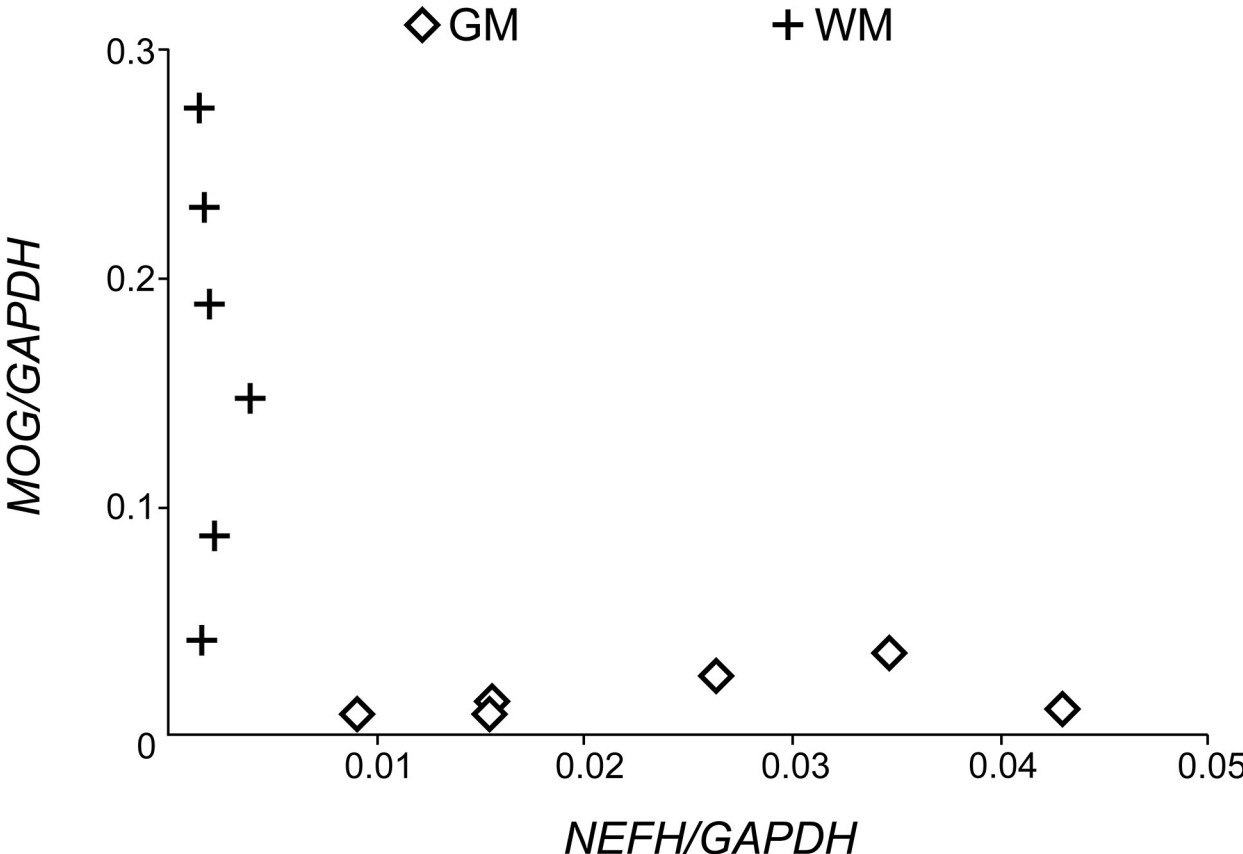

**Fig 2. Expression level of *NEFH* and *MOG* in the GM and WM.** Concentration of *NEFH* and *MOG* mRNA calculated by the $2^{-\Delta\Delta Ct}$ method, which was divided by that of *GAPDH*. Data are presented as the mean of triplicates. GM, grey matter; WM, white matter.

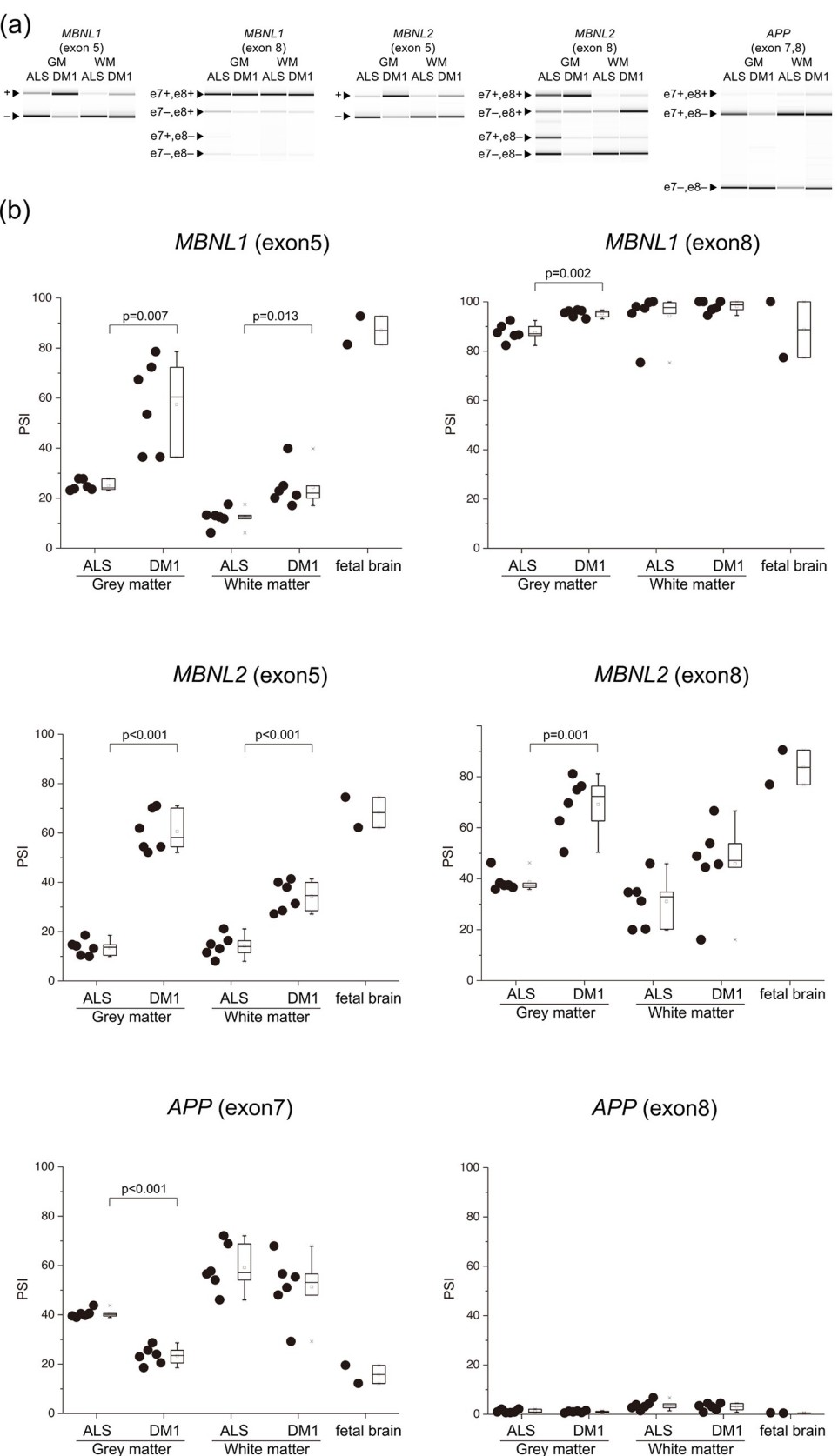

**Fig 3. Aberrant splicing between the GM and WM.** (a) Representative RT-PCR products from the frontal lobe GM and WM in ALS and DM1. (b) Inclusion ratios of splicing changes in the GM and WM. PSI values of all examined genes were compared by Welch's T-test. In the box-and-whisker plot, the line inside the box represents the median and the square symbol represents the average. ALS, amyotrophic lateral sclerosis; DM1, myotonic dystrophy type 1; GM, grey matter; WM, white matter.

functional consequences of such a minor difference might be minimal. It is difficult to draw a conclusion without further examination.

Analysis between the GM and WM revealed that, compared to the ALS, splicing changes in the GM of DM1 patients exceeded those in WM, in 7 out of the 15 examined exons. When several brain regions were compared, we collected RNA from both the GM and WM, and did not evaluate the ratio of GM to WM. Therefore, we assume that the distribution of PSI observed in the comparison among several brain regions may have been caused by the variation in the ratio of GM/WM in the collected samples.

This study revealed that the degree of mis-splicing in DM1 differs between the GM and WM. The GM is composed of the neuronal cell bodies, protoplasmic astrocytes, and microglial

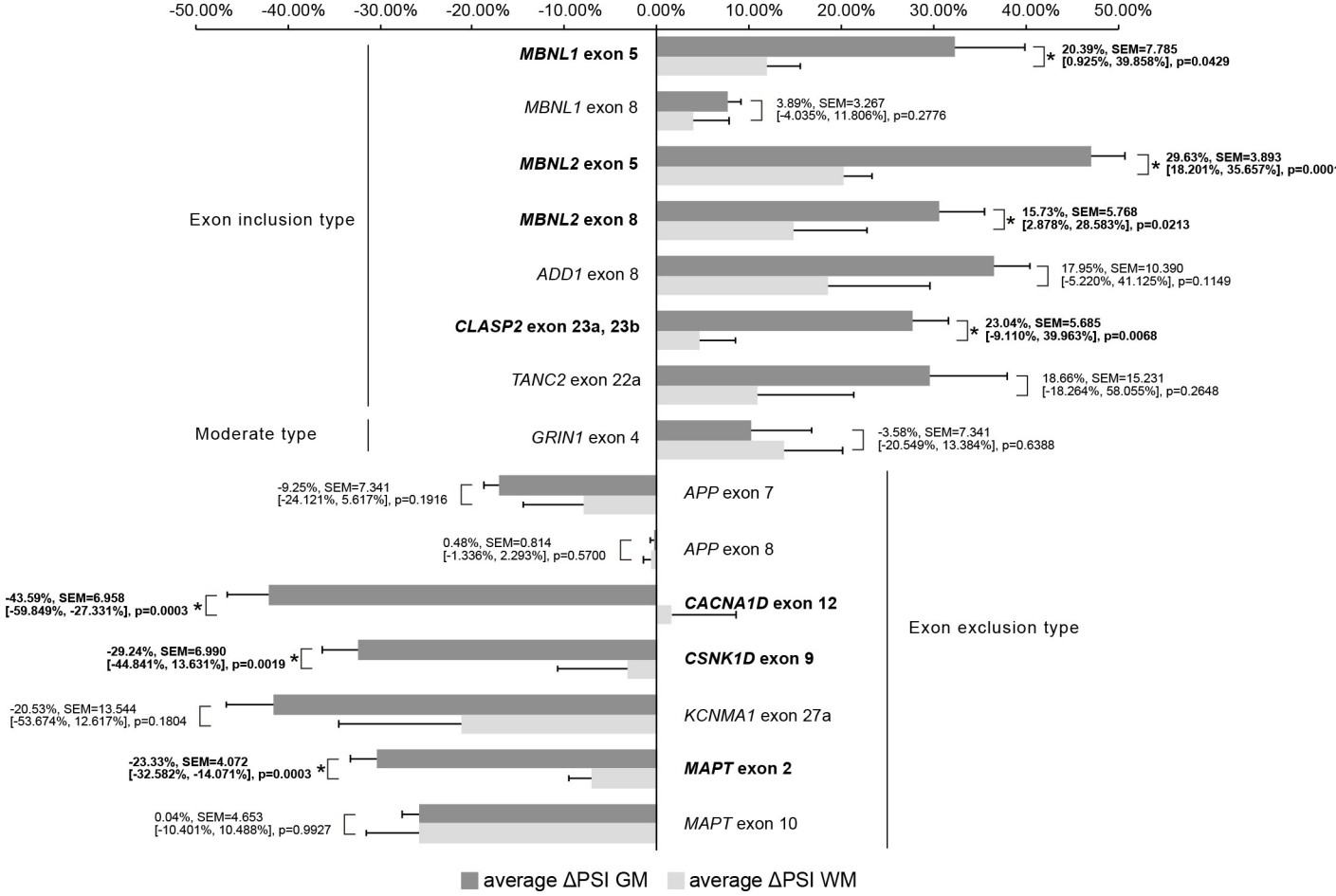

**Fig 4. Comparison of ΔPSI GM and ΔPSI WM.** This bar graph shows average ΔPSI GM and average ΔPSI WM. The error bars represent SEM. Average Δ PSI values of all examined genes were compared by Welch's T-test. The numbers in the graph indicate the mean difference, SEM, [Lower CL, Upper CL], and p-value in this order for comparison between average ΔPSI GM and average ΔPSI WM. The gene names and numbers with statistical significance were written in bold with asterisk. GM, grey matter; WM, white matter.

cells, while the WM comprises axons, oligodendrocytes, fibrous astrocytes, microglial cells, and ependymal cells (near the brain ventricle) [29]. In this study, as mRNA was extracted from the WM just under the GM, ependymal cells were not included in the samples. There are a few possibilities that could explain why these differences in splicing between the GM and WM occurred. One possibility is that these differences may be affected by the distribution of each splice isoform between the axon and neuronal cell body. The WM is less transcriptionally active than the GM, and mRNA in the WM may represent mRNA flow from the GM [13], since the mRNAs are transcribed in the neuronal cell body, and part of them are transported to the axon [30]. *MBNL1/2* exon 5 includes nuclear localization signals, and it is reported that the isoforms with this exon tend to be exclusively expressed in the nucleus [12]. Inclusion of exon 8 reduces the hairpin loop mobility of MBNL2 [12]. Since both exons tend to be present in DM1 patients, there is less MBNL in the cytoplasm. In order to target to specific compartments, MBNL binds to the distal 3'UTR protein binding sites and may intercede isoform specific mRNA localization [31]. Hence, it is possible that the presence or absence of each alternative exon may influence its binding ability to the 3'UTR of *MBNL* and the transport of its mRNA. Consequently, aberrant splicing of *MBNL* mRNA and the resulting intracellular localization shift of MBNL in DM1 patients might affect axonal transport of each spliced mRNA isoform. Another possibility is that splicing differences between the GM and WM may be influenced by splicing patterns of other cells such as oligodendrocytes, fibrous astrocytes, and microglial cells of the WM. Jiang showed that in DM1 patients, PSI for APP exon 7 were lower than in the control [10]. APP exon 7 and 8 are excluded from the neuronal APP isoform (APP 695), but are included in the astrocytic APP isoform (APP 751, 770) [32]. We showed that in the GM, PSI for APP exon 7 in the DM1 was significantly lower than in the ALS, but there was no statistical significance in WM. From this result, we presumed that the effect of aberrant splicing of astrocyte in the WM may be negligible. These distinct splicing changes may be due to differences in the number of CTG repeats in the cells between the GM and WM. Jinnai et al. examined the somatic instability of the CTG expansion in various regions of the CNS and found that the expansion in the GM was longer than that in the WM, although this difference did not reach statistical significance [33]. Possibly, this relatively short expansion in the WM might explain the modest level of aberrant splicing. In order to examine this possibility, it is necessary to isolate each cell type using cell culture or laser microdissection.

VBM and DWI tensor studies have shown that the effect of DM1 on the WM is more prominent than on the GM and that WM T2 hyperintensity lesions were found in the frontal, temporal (especially at the anterior temporal poles), and parietal lobes [21, 22]. However, this study showed more splicing defects in the GM than in the WM. It is unclear how these differences in the degree of splicing abnormalities relate to the predominant WM change. It remains unclear if the change in the WM is a result of Wallerian degeneration or a primary process of DM1 [17, 19]. As mentioned above, we suggested two possibilities explaining the differences in splicing defects between the GM and WM. Assuming the possibility that aberrant/fetal splicing mRNA isoforms are difficult to transport to the axon, aberrantly spliced isoforms would increase in the neuronal cell body, which could cause axonal injury as a consequence of Wallerian degeneration. The possibility that fewer splicing defects occurred in the myelin sheath of oligodendrocytes compared to the neuronal cell body does not explain the predominantly affected WM as shown by neuroimaging. Taken together, we hypothesize that WM lesions were caused by Wallerian degeneration due to neuronal cell body damage. Although there are limited findings on how each splice isoform protein is distributed, this is the first study showing the differences in splicing regulation and the degree of aberrant splicing between the GM and WM in DM1 patients.

## Limitations

This study had some limitations. First, we used ALS brain samples as the disease control. ALS tissues show splicing deregulation [34] and ALS brains may also display significant changes in WM and GM [35]. Our previous study showed statistical significance in the splicing defects in the temporal lobe of DM1 compared with either disease controls (7 out of 9 samples were from ALS) or healthy controls [4]. However, other areas, GM and WM of healthy controls have not been examined, and further study is needed in the future. Second, although it is important to determine whether the degree of mis-splicing, as the difference between the GM and WM, is related to each type of DM1 and the degree of cognitive decline, we could not determine a clear trend in this study, probably because the sample size was small and the variability of the PSI values between samples was large.

## Conclusion

In this study, we showed that splicing changes are relatively modest in the WM compared to the GM in the brain of DM1, which indicated the possibility that aberrant/fetal splicing isoforms may not be transported to the axon. Our result suggests that the predominant effects of DM1 on the WM might represent axonal injury by Wallerian degeneration due to GM dominant aberrant splicing. We believe that exploring the distribution of each spliced protein isoform by immunostaining can help in our understanding of the pathological mechanisms of DM1. Future studies should also investigate mRNA and protein transport and local translation in the axons of neuronal cells.

## Supporting information

**S1 Fig. MBNL1 exon 8 aberrant splicing among several regions of the brain.** Inclusion ratios of splicing changes in several brain regions. PSI values of MBNL1 exon 8 was compared by Welch's T-test. In box-and-whisker plot, the line inside the box is the median, square symbol is the average. ALS, amyotrophic lateral sclerosis; DM1, myotonic dystrophy type 1; Frontal., Frontal lobe; Temporal., Temporal lobe; Hippo., Hippocampus; Cerebel., Cerebellum.
(TIF)

**S2 Fig. Aberrant splicing between the GM and WM.** Inclusion ratios of splicing changes in the GM and WM. PSI values of all examined genes were compared by Welch's T-test. In box-and-whisker plot, the line inside the box is the median, square symbol is the average. ALS, amyotrophic lateral sclerosis; DM1, myotonic dystrophy type 1.
(TIF)

## Acknowledgments

The authors thank the Research Resource network Japan for the human brain samples and editage (www.editage.jp) for English language editing.

## Author Contributions

**Conceptualization:** Takashi Kimura, Harutoshi Fujimura, Kenji Jinnai, Hiroo Yoshikawa.

**Funding acquisition:** Takashi Kimura.

**Investigation:** Masamitsu Nishi, Mitsuru Furuta, Koichi Suenaga.

**Methodology:** Masataka Igeta.

**Resources:** Tsuyoshi Matsumura, Harutoshi Fujimura, Kenji Jinnai.

**Supervision:** Koichi Suenaga, Hiroo Yoshikawa.

**Writing – original draft:** Masamitsu Nishi.

**Writing – review & editing:** Takashi Kimura, Mitsuru Furuta, Tsuyoshi Matsumura, Hiroo Yoshikawa.

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
