## [Decision Letter · Decision Letter 0]

16 Nov 2019

PONE-D-19-29543

Differences in splicing defects between the grey and white matter in myotonic dystrophy type 1

PLOS ONE

Dear Dr. Kimura,

Thank you for submitting your manuscript to PLOS ONE. After careful consideration, we feel that it has merit but does not fully meet PLOS ONE’s publication criteria as it currently stands. Therefore, we invite you to submit a revised version of the manuscript that addresses the points raised during the review process.

We would appreciate receiving your revised manuscript by Dec 31 2019 11:59PM. To enhance the reproducibility of your results, we recommend that if applicable you deposit your laboratory protocols in protocols.io, where a protocol can be assigned its own identifier (DOI) such that it can be cited independently in the future. For instructions see: http://journals.plos.org/plosone/s/submission-guidelines#loc-laboratory-protocols

We look forward to receiving your revised manuscript.

Kind regards,

Ruben Artero, Ph.D.

Academic Editor

PLOS ONE

Journal Requirements:

Additional Editor Comments:

The manuscript is of potential interest to the journal but the results presented do not fully support the conclusions, particularly, the main claim that missplicing severity differs between white and grey matter. A full description of the clinical samples is also missing and additional examples of missplicing should be included. Despite of great relevance, a merely descriptive paper is acceptable for Plos one, hence the characterization of the disease expansion size in tissue samples by small pool PCR is not a requisite to submit a revised version of the manuscript.

Reviewers' comments:

Reviewer's Responses to Questions

**Comments to the Author**

1. Is the manuscript technically sound, and do the data support the conclusions?

Reviewer #1: No

Reviewer #2: Partly

2. Has the statistical analysis been performed appropriately and rigorously? 

Reviewer #1: No

Reviewer #2: Yes

3. Have the authors made all data underlying the findings in their manuscript fully available?

Reviewer #1: Yes

Reviewer #2: Yes

4. Is the manuscript presented in an intelligible fashion and written in standard English?

Reviewer #1: Yes

Reviewer #2: Yes

5. Review Comments to the Author

Reviewer #1: This manuscript addresses an important question in the field of myotonic dystrophy brain disease: the characterisation of pathology distribution between different brain areas and between grey and white matter. The authors used standard and suitable techniques to tackle this problem, but the paper presents some critical flaws that weaken the robustness of the data and the conclusions.

Importantly, the authors do not provide statistical evidence to support differences in the extent of missplicing between GM and WM in DM1 brains. The analysis is purely descriptive, without visual support and appropriate statistical testing. This is critical, since it is the main message of the paper, even highlighted in the title. As a result, the final discussion and conclusion are exceedingly speculative, with little experimental support to some of the hypothesis put forward by the authors.

Abstract:

The abstract fails to summarise some important factual data reported by the authors. Instead the text is too speculative. It requires some careful re-writing.

Factual mistakes:

The manuscript includes a few factual mistakes that must be corrected:

• Page 5, line 87: “Aberrant splicing of MBNL1 in the DM1 brain has been previously reported [23], but that of MBNL2 is still unknown.”

• Page 16, page 274: “however, we are the first to demonstrate MBNL2 exon 5 and exon 8 inclusion in the DM1 brain.”

MBNL2 missplicing in human brains has been previously reported in at least a couple of publications (Hernandez-Hernandez, 2013, Rare Diseases, 1: e25553; Sicot, 2017, Cell Reports, 19: 2718).

Statistical methods:

• The selection of the statistical tests is not convincingly justified. The Welch’s t-test used assumes normal distribution. Have the authors tested the normality of their data prior to the test?

• In Figure 1 and Figure 3 the authors use a pairwise Welch’s t-test. It is unclear what has been “paired” in the analysis.

• Welch’s t-test is known to perform poorly with low samples sizes. The authors should consider a more conservative non-parametric test.

Exon nomenclature:

The nomenclature of exons is vague and ambiguous, giving ample opportunity to confusion and misunderstanding. The discrepancy between databases makes it impossible today to identify an exon simply by their number, without reporting the reference genome/database used. To unequivocally identify the alternative exons investigated, the authors should provide their size and, ideally, their genomic coordinates.

Experimental conditions and rationale:

• Quantitative analysis of alternative splicing requires linear conditions of PCR amplification. The number of cycles used (36 and 42) seems exceptionally high. One wonders if they fall within the linear range of amplification. Given the extensive comparisons performed and the small magnitude of some of the differences reported, this is a critical point. Adequate controls must be included to demonstrate the linearity of the PCR amplification.

• The rationale behind some experiments is not formulated and the implications of the conclusions are not properly outlined in the results section. E.g. western blot analysis of MBNL2 (page 15, lines 251 – 253). The relevance of this experiment is unknown, and the conclusions drawn are missing in the results section. Additionally, the variability between samples is very high making it very difficult to produce robust conclusions from the investigation of 3 patients and non-DM controls.

• There is an unjustified focus on the analysis of MBNL2 protein. In reality we do not know the distribution of MBNL1 and MBNL2 between GM and WM. The parallel analysis of MBNL1 protein levels (and eventually CELF proteins) would provide some useful insight and support to some of the hypotheses discussed in the final section of the paper.

Data presentation and discussion:

• The text in page 13 (lines 219 – 222) does not match the data presented in Figure 3. The authors should review the inclusion and exclusion of some alternative cassettes.

• The data discussed on pages 13-14 (lines 230 – 236) is difficult to follow without a visual support (table or figure). More importantly, this data requires statistically testing to be convincing and to support the title of the paper.

Typographic errors:

Overall the quality of the English is good, but there are a few minor typographic errors. E.g. Page 5, line 93: “…and the other genes that controlled by MBNL2”. The authors should carefully review their manuscript.

Reviewer #2: Myotonic dystrophy is principally a muscle disease. The brain is also affected and a mis-splicing of several targets has been described. In contrast to muscle in which MBNL1 is the major paralogous expressed, MBNL1 and MBNL2 are expressed in the brain suggesting that the loss of one or both is needed to produce the mis-splicing events. However, this is without knowing how MBNL1 and MBNL2 are expressed in the brain tissue and what is their distribution between white and grey matter. The present work is interesting and need further experiments for being conclusive.

Major concerns:

1) Nothing is known about the control and DM1 patients

age, gender, duration of the disease, congitive status, type of DM1 ...

2) It is mandatory to know the stutus of the mutation in the brain tissue analysed using small-pool PCR. Thjis will give the parental transmission, instability and min and maximal length of CUG expansion

3) make a correlation analysis between the mutation status and the intensity of the mis-splicing events

It will provide information about the sensivitity of the mis-splicing events with regards what is already described in the litterature

4) Other transcripts should be include such as APP, Cacna1d.... APP is very important because the APP isoform 770 and 751 are expressed in the glial cells and not in neurons and therefore a mis-splicing of APP would suggest that the mutation is affecting glial cells as well and this has been scarcely documented (Goodwin et al., 2015).

5) Poly A selection has been described for MBNL2 in mouse brain, is it also true for the human brain?

Minor concerns

Could authors use the numbering of exons as the appears in the litterature rather than as they appear in database. For instance, tau splicing is related to exon 2, 3 and 10 not to exon 12.

Comments should be considered otherwise it makes the present work overwall confirmatory.

6. PLOS authors have the option to publish the peer review history of their article (what does this mean?). If published, this will include your full peer review and any attached files.

Reviewer #1: No

Reviewer #2: Yes: Sergeant Nicolas

---

## [Author Response · Author response to Decision Letter 0]

31 Jan 2020

Jan 31, 2020

Dr. Joerg Heber

Editor-in-Chief

PLOS ONE

Dear Editor:

Thank you for inviting us to submit a revised draft of our manuscript entitled, “Differences in splicing defects between grey and white matter in myotonic dystrophy type 1 patients.” to PLOS ONE. We also appreciate the time and effort you and each of the reviewers have dedicated to providing insightful feedback on ways to strengthen our paper. Thus, it is with great pleasure that we resubmit our article for further consideration. We have incorporated changes that reflect the detailed suggestions you have graciously provided. We also hope that our edits and the responses we provide below satisfactorily address all the issues and concerns you and the reviewers have noted.

To facilitate your review of our revisions, the following is a point-by-point response to the questions and comments delivered in your letter dated Nov 16 2019.

The paper was coauthored by Masamitsu Nishi, Masataka Igeta, Mitsuru Furuta, Koichi Suenaga, Tsuyoshi Matsumura, Harutoshi Fujimura, Kenji Jinnai, and Hiroo Yoshikawa. In this draft of our manuscript we added Dr. Igeta as a coauthor as he has newly contributed to the statistical analysis.

According to editor’s recomendation, we provided the original underlying images for all blot data at http://www2.hyo-med.ac.jp/~ma-nishi/ .

 

Reviewer #1

Response: We thank the reviewer for his or her thoughtful and thorough review and believe the input has made our research more scientifically meaningful. We address all of the concerns here.

 . 

Abstract:

The abstract fails to summarise some important factual data reported by the authors. Instead the text is too speculative. It requires some careful re-writing.

→The manuscript was reconsidered and refined.

Factual mistakes:

The manuscript includes a few factual mistakes that must be corrected:

• Page 5, line 87: “Aberrant splicing of MBNL1 in the DM1 brain has been previously reported [23], but that of MBNL2 is still unknown.”

• Page 16, page 274: “however, we are the first to demonstrate MBNL2 exon 5 and exon 8 inclusion in the DM1 brain.”

MBNL2 missplicing in human brains has been previously reported in at least a couple of publications (Hernandez-Hernandez, 2013, Rare Diseases, 1: e25553; Sicot, 2017, Cell Reports, 19: 2718).

→ I modified the manuscript that you pointed out, and added these papers in the reference. Missplicing of MBNL2 exon 5 (54nt) in human brains has been reported, but that of MBNL2 exon 8 is not reported. 

Statistical methods:

• The selection of the statistical tests is not convincingly justified. The Welch’s t-test used assumes normal distribution. Have the authors tested the normality of their data prior to the test?

→We asked Dr. Igeta, a medical statistician of our college to review the statistical method and decided to include him as a co-author.

We performed the Shapiro-Wilk test for testing the normality. If the Shapiro-Wilk test was significant at least one group within a group comparison, we applied the Wilcoxon rank sum test. Then we observed the consistent results of the Welch’s t-test and Wilcoxon test for these cases.

• In Figure 1 and Figure 3 the authors use a pairwise Welch’s t-test. It is unclear what has been “paired” in the analysis.

→ “Pairwise” is a pair of the control and DM1 at each brain areas. It is also described in the manuscript. The test between the GM and WM in the control was omitted from the present results. Figure 3 is modified accordingly.

• Welch’s t-test is known to perform poorly with low samples sizes. The authors should consider a more conservative non-parametric test.

→The Wilcoxon rank sum test evaluates the location shift between groups assuming the same shape of data distributions. We considered the assumption of the Wilcoxon rank sum test does not necessarily fit to our data. For example, the PSI of MBNL1 (exon 5) at Hippocampus in Figure1 have different variances between groups, which indicate difference shape of distributions. We can see many of our data shows such distributions. Therefore, we showed the p^ values of the Welch’s t-test because there is no statistical test uniformly recommended to our data.

Exon nomenclature:

The nomenclature of exons is vague and ambiguous, giving ample opportunity to confusion and misunderstanding. The discrepancy between databases makes it impossible today to identify an exon simply by their number, without reporting the reference genome/database used. To unequivocally identify the alternative exons investigated, the authors should provide their size and, ideally, their genomic coordinates.

→We corrected MAPT exon number and added number of base pairs of all examined exons.

Experimental conditions and rationale:

• Quantitative analysis of alternative splicing requires linear conditions of PCR amplification. The number of cycles used (36 and 42) seems exceptionally high. One wonders if they fall within the linear range of amplification. Given the extensive comparisons performed and the small magnitude of some of the differences reported, this is a critical point. Adequate controls must be included to demonstrate the linearity of the PCR amplification.

→To determine the cycle number, we confirmed the linearity of the PCR amplification at 32 (for comparison among several brain regions) or 36 (for comparison between the GM and WM). Then we retried RT-PCR study at each cycle.

• The rationale behind some experiments is not formulated and the implications of the conclusions are not properly outlined in the results section. E.g. western blot analysis of MBNL2 (page 15, lines 251 – 253). The relevance of this experiment is unknown, and the conclusions drawn are missing in the results section. Additionally, the variability between samples is very high making it very difficult to produce robust conclusions from the investigation of 3 patients and non-DM controls.

• There is an unjustified focus on the analysis of MBNL2 protein. In reality we do not know the distribution of MBNL1 and MBNL2 between GM and WM. The parallel analysis of MBNL1 protein levels (and eventually CELF proteins) would provide some useful insight and support to some of the hypotheses discussed in the final section of the paper.

→ As you pointed out, we decided to omit the protein analysis from this post due to the problem of the number of sample.

Data presentation and discussion:

• The text in page 13 (lines 219 – 222) does not match the data presented in Figure 3. The authors should review the inclusion and exclusion of some alternative cassettes.

→We corrected the errors.

• The data discussed on pages 13-14 (lines 230 – 236) is difficult to follow without a visual support (table or figure). More importantly, this data requires statistically testing to be convincing and to support the title of the paper.

→We added a figure showing the average level of PSI change and Statistical tests were also performed (Figure 4).

Typographic errors:

Overall the quality of the English is good, but there are a few minor typographic errors. E.g. Page 5, line 93: “…and the other genes that controlled by MBNL2”. The authors should carefully review their manuscript.

→We corrected the typographic errors.

 

Reviewer #2: 

Response: We would like to thank the reviewer for his thoughtful comments and efforts towards improving our manuscript. We address all of the concerns here.

Major concerns:

1) Nothing is known about the control and DM1 patients

age, gender, duration of the disease, congitive status, type of DM1 ...

→The age, gender, duration, and cognitive status of each sample were summarized in table 2.

2) It is mandatory to know the stutus of the mutation in the brain tissue analysed using small-pool PCR. This will give the parental transmission, instability and min and maximal length of CUG expansion

3) make a correlation analysis between the mutation status and the intensity of the mis-splicing events

It will provide information about the sensivitity of the mis-splicing events with regards what is already described in the literature

→From personal communication with Dr Nakamori, Osaka University, we know that the analysis of small pool PCR using brain samples is technically difficult. 　In addition, the editor suggested that “the characterization of the disease expansion size in tissue samples by small pool PCR is not a requisite to submit a revised version of the manuscript.” Taken together, we have determined not to carry out the analysis.

4) Other transcripts should be include such as APP, Cacna1d.... APP is very important because the APP isoform 770 and 751 are expressed in the glial cells and not in neurons and therefore a mis-splicing of APP would suggest that the mutation is affecting glial cells as well and this has been scarcely documented (Goodwin et al., 2015).

→ We confirmed that the isoform changes depending on the presence or absence of exon7 and 8 in APP. Then we made primers for exon 7 and 8 splicing and analyzed for APP splicing between GM and WM.

5) Poly A selection has been described for MBNL2 in mouse brain, is it also true for the human brain?

→We didn’t choose Poly A selection in this study.

Minor concerns

Could authors use the numbering of exons as the appears in the litterature rather than as they appear in database. For instance, tau splicing is related to exon 2, 3 and 10 not to exon 12.

→I corrected MAPT exon number and added number of base pairs of all examined exons.

---

## [Decision Letter · Decision Letter 1]

27 Feb 2020

PONE-D-19-29543R1

Differences in splicing defects between the grey and white matter in myotonic dystrophy type 1 patients

PLOS ONE

Dear Dr. Kimura,

Thank you for submitting your manuscript to PLOS ONE. After careful consideration, we feel that it has merit but does not fully meet PLOS ONE’s publication criteria as it currently stands. Therefore, we invite you to submit a revised version of the manuscript that addresses the points raised during the review process.

Some of the points raised by the reviewers were well addressed and, overall, the paper is now less speculative.

However, some important weaknesses still remain and they raise important reservations. Importantly, both reviewers still agree that the paper is only partly sound, which is critical for acceptance. The use of ALS patients as controls challenges the robustness of the data and the statistics are still not clear. Please make every effort to address all comments raised by the reviewers.

We would appreciate receiving your revised manuscript by Apr 12 2020 11:59PM. To enhance the reproducibility of your results, we recommend that if applicable you deposit your laboratory protocols in protocols.io, where a protocol can be assigned its own identifier (DOI) such that it can be cited independently in the future. For instructions see: http://journals.plos.org/plosone/s/submission-guidelines#loc-laboratory-protocols

We look forward to receiving your revised manuscript.

Kind regards,

Ruben Artero, Ph.D.

Academic Editor

PLOS ONE

Reviewers' comments:

Reviewer's Responses to Questions

**Comments to the Author**

1. If the authors have adequately addressed your comments raised in a previous round of review and you feel that this manuscript is now acceptable for publication, you may indicate that here to bypass the “Comments to the Author” section, enter your conflict of interest statement in the “Confidential to Editor” section, and submit your "Accept" recommendation.

Reviewer #1: (No Response)

Reviewer #2: All comments have been addressed

2. Is the manuscript technically sound, and do the data support the conclusions?

Reviewer #1: Partly

Reviewer #2: Partly

3. Has the statistical analysis been performed appropriately and rigorously? 

Reviewer #1: No

Reviewer #2: Yes

4. Have the authors made all data underlying the findings in their manuscript fully available?

Reviewer #1: Yes

Reviewer #2: Yes

5. Is the manuscript presented in an intelligible fashion and written in standard English?

Reviewer #1: Yes

Reviewer #2: Yes

6. Review Comments to the Author

Reviewer #1: I commend the authors for the changes introduced in the revised version of the manuscript, and for their efforts to strengthen the quality of their work and the clarity of this report. As previously highlighted, the results reported by Kimura and colleagues are interesting. However, I still have some concerns that have not been convincingly resolved by the revision.

Abstract

The second sentence is oversimplified and it gives the wrong idea that DM1 pathogenesis is explained by MBNL2 sequestration alone. We know that MBNL1 is also sequestered by RNA foci and plays a critical role, notably in the brain (Jiang et al. 2004; Hernandez-Hernandez et al. 2013, to cite but a few). This sentence should be modified.

Introduction

Line 4. “It has been reported that fetal splice isoforms increase in adult DM1 tissues as a result of MBNL1/2 deletion”.

“Deletion” is not the most suitable word in this context. “Depletion” or “sequestration” would be more appropriate to describe the biological situation accurately.

Materials and methods

I am concerned that the ALS control samples used in this study may not be the most suitable to perform a careful comparison with DM1 brains. ALS tissue not only shows splicing deregulation [Nussbacher et al (2019) Neuron, 102: 294; Butti et al (2019) Frontiers in Genetics, 9: 712], but ALS brains may also display significant changes in white and grey matter [Zhou et al (2017) Mol Med Rep, 16: 4379; Turner et al (2015) Curr Neurol Neurosci Rep, 15: 45]. The authors should carefully address this point in their manuscript, and acknowledge its implications on the interpretation of results.

I think there is some sort of confusion in the use of paired and unpaired statistical tests. In Figure 1, when comparing PSI between control and DM1 samples, the authors claim to have used a paired Welch’s t-test. I do not understand the criteria used to pair subjects and I strongly believe this makes no biological sense. In my opinion the only situation where we could apply a paired Welch’s t-test would be the comparison of DPSI between GM and WM in Figure 4, in which we must pair GM and WM samples collected from the same individual. Surprisingly, legend of Figure 4 does not refer to the use of a paired statistical test.

Results:

Figure 1 and Figure 3. The inclusion of the electrophoretic profiles is very useful and illustrates well the splicing defects reported in the graphs. However, it is unclear which exons are represented by each band. Proper labeling should be included, as it was done in the case of APP (exons 7 and 8) in Figure 3. The labelling is particularly important and relevant when multiple bands are detected for the same gene (more than 2), like in the analysis of MBNL2 (exon 8).

Page 14, line 225.

The authors write: “confirmed that the GM and WM were correctly separated”.

I believe this is an overstatement, and authors should tone it down the text, simply referring that samples were “enriched” for GM and WM.

Main text, page 14, line 226-227.

I find it hard to include TANC2 in the group of exon inclusion type in the fetal brain, since PSI for this alternative exon in this tissue is around 50% (if not slightly below). Are the authors referring to the adult brain? The sentence is vague, and the criteria to define exon inclusion version exon exclusion are not clearly defined.

Figure 3 and main text, page 14 lines 229-232.

The values in the text and the graph figure for MBNL1 (exon 5) do not match. In the text we read that PSI for MBNL1 exon 5 in the GM of DM1 patients is 57.43%. If we examine the graph carefully, the mean PSI represented is above 60% for the same tissue in the same group of subjects. The authors should correct this problem and review all the graphs carefully.

Figure 4.

The p values of some comparisons are not included in the figure, so one assumes the difference is not statistically different. E.g. TANC2 exon 22a. However, it is confusing that simply by looking at the graph, the average DPSI represented for grey and white matter appears to be considerably different. To overcome this apparent inconsistency the authors should either represent SEM or plot the individual values around the average calculated.

Discussion

Page 17, lines 281-289.

The authors elaborate on the splicing defects of GRIN1 in the hippocampus. Although interesting, two considerations should be highlighted in the discussion. The extent of the splicing defect relative to ALS patients (in other words, the value of DPSI) is very low, and the functional consequences of such a minor difference might also be minimal. Regardless of this limitation, the splicing defect of GRIN1 in the hippocampus does not represent a return to an embryonic splicing profile, and this is very interesting per se and should be further stressed.

Reviewer #2: In this revised manuscript authors has addressed almost completely all concerns except for the size of CTG expansion. Thus, their diagnosis were mainly based on the disease onset.

However, as now indicated in table DM1 patient include early, juvenile, adult and late onset cases. Even if the number of sample is low, it would have been of interest to determine whether the amplitude of mis-splicing as the difference between grey and white matter is observed in each sub-type of the disease. This is in the scope of this paper since the mechanism underlying the mis-splicing is different between early or juvenile and adult / late onset. Large expansion are transmitted in the early and juvenile form whereas large expansion are acquired with ageing in adult and late onset DM1 case.

More importantly, authors have referred ALS brain tissue as a "control" tissue. The cause of ALS is not known and it is well-known that a mis-transport of ARN is instrumental of ALS. Please refer to ALS rather than to control brain tissue.

The referee has no other comment

7. PLOS authors have the option to publish the peer review history of their article (what does this mean?). If published, this will include your full peer review and any attached files.

Reviewer #1: No

Reviewer #2: No

---

## [Author Response · Author response to Decision Letter 1]

2 Apr 2020

Response to reviewers

Reviewer #1: I commend the authors for the changes introduced in the revised version of the manuscript, and for their efforts to strengthen the quality of their work and the clarity of this report. As previously highlighted, the results reported by Kimura and colleagues are interesting. However, I still have some concerns that have not been convincingly resolved by the revision.

>> We thank again the reviewer for his or her thoughtful and thorough review and believe the input has made our research more scientifically meaningful. We address all of the concerns here. 

Abstract

The second sentence is oversimplified and it gives the wrong idea that DM1 pathogenesis is explained by MBNL2 sequestration alone. We know that MBNL1 is also sequestered by RNA foci and plays a critical role, notably in the brain (Jiang et al. 2004; Hernandez-Hernandez et al. 2013, to cite but a few). This sentence should be modified.

>> Thank you for your suggestion. We have incorporated your comments by changing MBNL2 to MBNL1 / 2 in the abstract.

Introduction

Line 4. “It has been reported that fetal splice isoforms increase in adult DM1 tissues as a result of MBNL1/2 deletion”.

“Deletion” is not the most suitable word in this context. “Depletion” or “sequestration” would be more appropriate to describe the biological situation accurately.

>>We have reflected this comment by correcting the word.

Materials and methods

I am concerned that the ALS control samples used in this study may not be the most suitable to perform a careful comparison with DM1 brains. ALS tissue not only shows splicing deregulation [Nussbacher et al (2019) Neuron, 102: 294; Butti et al (2019) Frontiers in Genetics, 9: 712], but ALS brains may also display significant changes in white and grey matter [Zhou et al (2017) Mol Med Rep, 16: 4379; Turner et al (2015) Curr Neurol Neurosci Rep, 15: 45]. The authors should carefully address this point in their manuscript, and acknowledge its implications on the interpretation of results.

>>Thank you for providing these insights. The "control" in the text has been corrected to “ALS”. In addition, limitations paragraph was set up to respond to the concerns using ALS as disease control.

I think there is some sort of confusion in the use of paired and unpaired statistical tests. In Figure 1, when comparing PSI between control and DM1 samples, the authors claim to have used a paired Welch’s t-test. I do not understand the criteria used to pair subjects and I strongly believe this makes no biological sense. In my opinion the only situation where we could apply a paired Welch’s t-test would be the comparison of DPSI between GM and WM in Figure 4, in which we must pair GM and WM samples collected from the same individual. Surprisingly, legend of Figure 4 does not refer to the use of a paired statistical test.

>>This is an important point. We deleted the term, “paired”, and its related description in the footnote of Figure 1 for correction. The statistical test which was actually applied in Figure1 was the (unpaired) Welch’s t-test as described in the section on “Statistical analysis”. Regarding to this correction, the footnotes of Figure 3, S1 and S2 were modified as well.

On the other hand, the footnote of Figure 4 related to the (unpaired) Welch’s t-test was not modified. We would like to explain more detail here about the statistical analysis for Figure 4, which was described as “The p-values for the difference between average ΔPSI GM and average ΔPSI WM ...” in the section on “Statistical analysis”. The definition of ΔPSI is “the average level of PSI change (ΔPSI), which was obtained by subtracting the average of PSI of the ALS from the DM1” as shown in the section on “Statistical analysis”. For example, ΔPSI GM are calculated as the difference of the averages of PSI of GM between groups (DM1-ALS). Furthermore, since there is no missing data in our PSI data, ΔPSI GM -ΔPSI WM, which was the basis for calculating the p-value in Figure 4, can be converted as follows:

ΔPSI GM - ΔPSI WM

= (average PSI of DM1 of GM - average PSI of ALS of GM) 

- (average PSI of DM1 of WM - average PSI of ALS of WM) 

= (average of PSI of (GM-WM) of DM1 - average PSI of (GM-WM) of ALS)

(GM-WM) in the above equation is a difference within a subject. However, the whole last equation expresses a comparison between DM1 and ALS groups. Therefore, we calculated the p-values in Figure 4 based on the (unpaired) Welch’s t-test for testing the difference between “average of PSI of (GM-WM) of DM1” and “average PSI of (GM-WM) of ALS” in the above equation.

Results:

Figure 1 and Figure 3. The inclusion of the electrophoretic profiles is very useful and illustrates well the splicing defects reported in the graphs. However, it is unclear which exons are represented by each band. Proper labeling should be included, as it was done in the case of APP (exons 7 and 8) in Figure 3. The labelling is particularly important and relevant when multiple bands are detected for the same gene (more than 2), like in the analysis of MBNL2 (exon 8).

>>We added the labels for exons7 and 8 of MBNL1 and MBNL2 in the figure 1.

Page 14, line 225.

The authors write: “confirmed that the GM and WM were correctly separated”.

I believe this is an overstatement, and authors should tone it down the text, simply referring that samples were “enriched” for GM and WM.

>>Thank you for your suggestion. We corrected the manuscript as you pointed out.

Main text, page 14, line 226-227.

I find it hard to include TANC2 in the group of exon inclusion type in the fetal brain, since PSI for this alternative exon in this tissue is around 50% (if not slightly below). Are the authors referring to the adult brain? The sentence is vague, and the criteria to define exon inclusion version exon exclusion are not clearly defined.

>>We agree with that TANC2 cannot be included in either pattern. We changed the patterns of AS in the fetal brain to three: 1) exon inclusion type (PSI ≥ 60%, ADD1, CLASP2, MBNL1/2), 2) exon exclusion type (PSI < 40%, APP, CACNA1D, CSNK1D, GRIN1, KCNMA1, MAPT) (Fig 3, S2 Fig), and 3) moderate type (40% ≤ PSI < 60%, TANC2). The manuscript and figures have been revised accordingly.

Figure 3 and main text, page 14 lines 229-232.

The values in the text and the graph figure for MBNL1 (exon 5) do not match. In the text we read that PSI for MBNL1 exon 5 in the GM of DM1 patients is 57.43%. If we examine the graph carefully, the mean PSI represented is above 60% for the same tissue in the same group of subjects. The authors should correct this problem and review all the graphs carefully.

>>You have raised an important query. In our box-and-whisker plot, the line inside the box is the median, square symbol is the average. We believe that the graph has correctly shown that the average PSI for MBNL1 exon5 in the GM of DM1 patients is 57.43%. To make this point clear, we added the sentence in the figure legends explaining that the average value is represented by square symbol in the box-and-whisker plot.

Figure 4.

The p values of some comparisons are not included in the figure, so one assumes the difference is not statistically different. E.g. TANC2 exon 22a. However, it is confusing that simply by looking at the graph, the average DPSI represented for grey and white matter appears to be considerably different. To overcome this apparent inconsistency the authors should either represent SEM or plot the individual values around the average calculated.

>>This is an important perspective. We added summary statistics such as SEM and 95% confidence interval in Figure 4.

Discussion

Page 17, lines 281-289.

The authors elaborate on the splicing defects of GRIN1 in the hippocampus. Although interesting, two considerations should be highlighted in the discussion. The extent of the splicing defect relative to ALS patients (in other words, the value of DPSI) is very low, and the functional consequences of such a minor difference might also be minimal. Regardless of this limitation, the splicing defect of GRIN1 in the hippocampus does not represent a return to an embryonic splicing profile, and this is very interesting per se and should be further stressed.

>> Thank you for your suggestions. We have inserted sentences that emphasizes the two considerations you pointed out.

Reviewer #2: In this revised manuscript authors has addressed almost completely all concerns except for the size of CTG expansion. Thus, their diagnosis were mainly based on the disease onset.

>> We would like to thank again the reviewer for his thoughtful comments and efforts towards improving our manuscript. We address all of the concerns here. 

However, as now indicated in table DM1 patient include early, juvenile, adult and late onset cases. Even if the number of sample is low, it would have been of interest to determine whether the amplitude of mis-splicing as the difference between grey and white matter is observed in each sub-type of the disease. This is in the scope of this paper since the mechanism underlying the mis-splicing is different between early or juvenile and adult / late onset. Large expansion are transmitted in the early and juvenile form whereas large expansion are acquired with ageing in adult and late onset DM1 case.

>>Thank you for providing these insights. We agree with that it is important to determine whether the degree of mis-splicing, as the difference between the GM and WM, is related to each type of DM1 and the degree of cognitive decline. However, we could not determine a clear trend in this study, probably because the sample size was small and the variability of the PSI values between samples was large. We mentioned this point in the paragraph of Limitations.

More importantly, authors have referred ALS brain tissue as a "control" tissue. The cause of ALS is not known and it is well-known that a mis-transport of ARN is instrumental of ALS. Please refer to ALS rather than to control brain tissue.

>>Thank you for your suggestion. The "control" in the text has been corrected to “ALS”. In addition, limitations paragraph was set up to respond to the concerns using ALS as disease control.

---

## [Decision Letter · Decision Letter 2]

10 Apr 2020

PONE-D-19-29543R2

Differences in splicing defects between the grey and white matter in myotonic dystrophy type 1 patients

PLOS ONE

Dear Dr. Kimura,

Thank you for submitting your manuscript to PLOS ONE. After careful consideration, we feel that it has merit but does not fully meet PLOS ONE’s publication criteria as it currently stands. Therefore, we invite you to submit a revised version of the manuscript that addresses the points raised during the review process.

Thanks for the careful implementation of all reviewer comments. Before final acceptance, please consider the minor editing suggestions and additional statistical comparisons that the reviewer suggest because they may strengthen the conclusions of the paper.  

We would appreciate receiving your revised manuscript by May 25 2020 11:59PM. To enhance the reproducibility of your results, we recommend that if applicable you deposit your laboratory protocols in protocols.io, where a protocol can be assigned its own identifier (DOI) such that it can be cited independently in the future. For instructions see: http://journals.plos.org/plosone/s/submission-guidelines#loc-laboratory-protocols

We look forward to receiving your revised manuscript.

Kind regards,

Ruben Artero, Ph.D.

Academic Editor

PLOS ONE

Reviewers' comments:

Reviewer's Responses to Questions

**Comments to the Author**

1. If the authors have adequately addressed your comments raised in a previous round of review and you feel that this manuscript is now acceptable for publication, you may indicate that here to bypass the “Comments to the Author” section, enter your conflict of interest statement in the “Confidential to Editor” section, and submit your "Accept" recommendation.

Reviewer #1: All comments have been addressed

2. Is the manuscript technically sound, and do the data support the conclusions?

Reviewer #1: Yes

3. Has the statistical analysis been performed appropriately and rigorously? 

Reviewer #1: Yes

4. Have the authors made all data underlying the findings in their manuscript fully available?

Reviewer #1: Yes

5. Is the manuscript presented in an intelligible fashion and written in standard English?

Reviewer #1: Yes

6. Review Comments to the Author

Reviewer #1: The authors have done a very good job clarifying some of the points, and strengthening the quality and impact of their manuscript.

I would like to point out a couple of minor details and bring up point for discussion.

1. Introduction, line 51-52.

It should rather be: “… such as muscleblind-like (MBNL) proteins”

2. Materials and Methods, line 140.

As discussed, and changed elsewhere in the manuscript, I would suggest: “To ensure enrichment of the GM and WM, …”

3. Figure 4.

Would it be possible to introduce error bars directly on the graph, representing the SEM values? The change would certainly make the figure clearer.

Figure 4 (suggestion for the authors’ consideration).

The authors explain the rationale behind the statistics performed, and the selection of the unpaired Welch’s t-test. Point taken.

Biologically, it would make sense to compare the Delta-PSI of an exon between the GM and WM of the same individual. In other words, when looking at the entire patient set, a paired test should compare the Delta-PSI in GM and Delta-PSI in WM by matching the GM and WM PSI values for the same patient.

How to go about and calculate the Delta PSI per exon, per individual DM1 patient? The solution would be to use as reference value the mean/median PSI for each exon among the ALS disease controls.

My feeling is that such paired statistical analysis could bring up significant differences between GM and WM that are otherwise confounded by inter-individual variability.

I leave this point to the authors’ consideration.

7. PLOS authors have the option to publish the peer review history of their article (what does this mean?). If published, this will include your full peer review and any attached files.

Reviewer #1: No

---

## [Author Response · Author response to Decision Letter 2]

24 Apr 2020

Response to reviewers

Reviewer #1: The authors have done a very good job clarifying some of the points, and strengthening the quality and impact of their manuscript.

I would like to point out a couple of minor details and bring up point for discussion.

>> We would like to thank again the reviewer for his or her thoughtful comments and efforts towards improving our manuscript. We address all of the concerns here. 

1. Introduction, line 51-52.

It should rather be: “… such as muscleblind-like (MBNL) proteins”

>> We agree with you. We have corrected the manuscript accordingly.

2. Materials and Methods, line 140.

As discussed, and changed elsewhere in the manuscript, I would suggest: “To ensure enrichment of the GM and WM, …”

>> Thank you for your suggestion. We have modified the manuscript accordingly.

3. Figure 4.

Would it be possible to introduce error bars directly on the graph, representing the SEM values? The change would certainly make the figure clearer.

>>Thank you for your suggestion. We have added error bars representing the SEM values of each Delta PSI on Figure 4. 

4.Figure 4 (suggestion for the authors’ consideration).

The authors explain the rationale behind the statistics performed, and the selection of the unpaired Welch’s t-test. Point taken.

Biologically, it would make sense to compare the Delta-PSI of an exon between the GM and WM of the same individual. In other words, when looking at the entire patient set, a paired test should compare the Delta-PSI in GM and Delta-PSI in WM by matching the GM and WM PSI values for the same patient.

How to go about and calculate the Delta PSI per exon, per individual DM1 patient? The solution would be to use as reference value the mean/median PSI for each exon among the ALS disease controls.

My feeling is that such paired statistical analysis could bring up significant differences between GM and WM that are otherwise confounded by inter-individual variability.

I leave this point to the authors’ consideration.

>> Thank you for an interesting proposal about a paired statistical analysis. We agree that the paired test using reference value would provide more significant results. However, we have to recognize the paired analysis would require an important assumption: the adequacy on the reference value. Although the variance of PSI values in the ALS disease controls was smaller than that in DM1 patients, we think non-negligible variability was left in the mean or median PSI in the ALS subjects because it was calculated from only 6 subjects. Thus, we considered that it is difficult to explain the adequacy of taking the sample mean or median as the reference value. Therefore, we would like to hold our interpretation of the difference between GM and WM based on the current analyses.

---

## [Editor Report · Decision Letter 3]

27 Apr 2020

Differences in splicing defects between the grey and white matter in myotonic dystrophy type 1 patients

PONE-D-19-29543R3

Dear Dr. Kimura,

We are pleased to inform you that your manuscript has been judged scientifically suitable for publication and will be formally accepted for publication once it complies with all outstanding technical requirements.

With kind regards,

Ruben Artero, Ph.D.

Academic Editor

PLOS ONE
---

## [Editor Report · Acceptance letter]

4 May 2020

PONE-D-19-29543R3 

Differences in splicing defects between the grey and white matter in myotonic dystrophy type 1 patients 

Dear Dr. Kimura:

I am pleased to inform you that your manuscript has been deemed suitable for publication in PLOS ONE. Congratulations! Your manuscript is now with our production department. 

With kind regards,

on behalf of

Dr. Ruben Artero 

Academic Editor

PLOS ONE